# METALIC: META-LEARNING IN-CONTEXT WITH PROTEIN LANGUAGE MODELS

**Jacob Beck,**[*] **Shikha Surana, Manus McAuliffe, Oliver Bent, Thomas D. Barrett,**
**Juan Jose Garau Luis, & Paul Duckworth**
InstaDeep | `https://github.com/instadeepai/metalic`
Boston, MA, USA & London, UK

## ABSTRACT

Predicting the biophysical and functional properties of proteins is essential for *in silico* protein design. Machine learning has emerged as a promising technique for such prediction tasks. However, the relative scarcity of *in vitro* annotations means that these models often have little, or no, specific data on the desired fitness prediction task. As a result of limited data, protein language models (PLMs) are typically trained on general protein sequence modeling tasks, and then fine-tuned, or applied zero-shot, to protein fitness prediction. When no task data is available, the models make strong assumptions about the correlation between the protein sequence likelihood and fitness scores. In contrast, we propose meta-learning over a distribution of standard fitness prediction tasks, and demonstrate positive transfer to unseen fitness prediction tasks. Our method, called *Metalic* (Meta-Learning In-Context), uses in-context learning and fine-tuning, when data is available, to adapt to new tasks. Crucially, fine-tuning enables considerable generalization, even though it is not accounted for during meta-training. Our fine-tuned models achieve strong results with 18 times fewer parameters than state-of-the-art models. Moreover, our method sets a new state-of-the-art in low-data settings on ProteinGym, an established fitness-prediction benchmark. Due to data scarcity, we believe meta-learning will play a pivotal role in advancing protein engineering.

## 1 INTRODUCTION

The accurate prediction of functional and biophysical properties of proteins, collectively referred to here as fitness, is a critical challenge in the physical sciences with far-reaching implications for medical research, agriculture, and drug discovery. For example, fitness prediction can be used to optimize properties such as the binding affinity of a monoclonal antibody therapy to its target or the thermostability of enzymes functioning at high temperatures. While protein fitness can be measured *in vitro*, the process is laborious and time-consuming. Consequently, machine learning models have emerged as a powerful tool to predict fitness directly from amino acid sequences *in silico*. However, due to the complex, high-dimensional relationship between protein sequences and fitness, and the limited availability of high-quality data, accurate fitness prediction is a challenge.

Proteins can be encoded as sequences of characters representing amino acids, making protein language models (PLMs) effective for modeling them (Madani et al., 2020; Rives et al., 2021; Rao et al., 2021; Lin et al., 2022; Notin et al., 2023; Truong Jr & Bepler, 2024). By predicting masked amino acids, or subsequent amino acids, over known proteins at scale, PLMs can capture much of the structure and resultant properties deriving from the amino acid sequence. While PLMs are not directly trained to predict fitness, they are trained to model the likelihood of naturally occurring proteins, which has been found to correlate strongly with their fitness (Truong Jr & Bepler, 2024). In practice, large PLMs are fine-tuned on downstream protein fitness data for regression. While highly effective, PLMs provide limited utility given severely limited data, or no data. With limited data, learning the correct regression is difficult, even with informative pre-trained representations. With no data, it must be assumed that protein fitness is solely a function of protein likelihood (Truong Jr & Bepler, 2024; Hawkins-Hooker et al., 2024), and for masked language models, it is also assumed that each amino acid contributes independently to fitness (Meier et al., 2021; Notin et al., 2024).

While limited or no data may be available for specific protein fitness prediction tasks, there are often other tasks that can serve as a valuable source of guidance. For example, rather than assuming

---

[*]Jacob_Beck@alumni.brown.edu

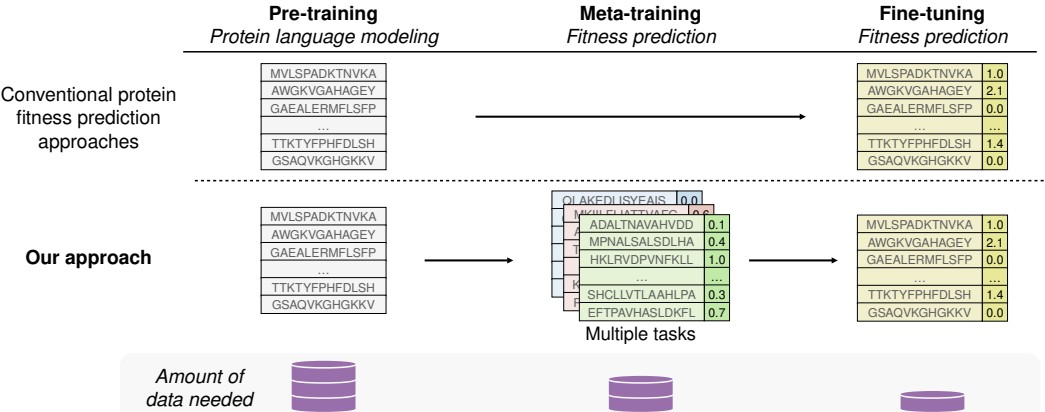

**Figure 1: An overview of meta-learning for protein fitness prediction.** PLMs are trained over massive quantities of unlabeled data. Using meta-learning, we also train over a smaller quantity of labelled fitness data. Using this extra data is critical given limited data for fine-tuning at test time.

that fitness is solely a function of protein likelihood, when no data is available for a given task, we can use data from other tasks to learn the relationship between PLM embeddings and fitness. Due to advances in high-throughput assays, such as deep mutational scanning Fowler & Fields (2014), data from other tasks is available to learn this relationship. One example is ProteinGym (Notin et al., 2024), which compiles over one hundred distinct fitness prediction tasks. Training over such a distribution of related tasks, to reason about new tasks, is referred to as *meta-learning* (Huisman et al., 2021; Hospedales et al., 2021; Beck et al., 2023a).

To address the challenge of limited data in protein fitness prediction, we propose *Metalic*, integrating in-context meta-learning, protein language models, and fine-tuning. *Metalic* builds on top of PLM embeddings and fine-tuning methods, but additionally, unlike baseline methods, adds a meta-learning phase over other protein prediction tasks. While some PLMs use in-context data (Rao et al., 2021; Notin et al., 2022; 2023; Truong Jr & Bepler, 2024), these methods do not *meta-learn* how to use their limited context for protein fitness prediction. The meta-learning phase is depicted in Fig. 1, and is critical to learning the relationship between PLM embeddings and fitness, given the limited fitness data available in each task at test time. While some meta-learning methods combine in-context learning and fine-tuning (Rusu et al., 2018; Vuorio et al., 2019), they do so by using computationally expensive higher-order gradients to account for fine-tuning during meta-learning. Crucially, *Metalic* leverages in-context meta-learning, and then subsequent fine-tuning, without accounting for fine-tuning during meta-learning. This novel combination is particularly computationally efficient, and enables *Metalic* to outperform more complicated methods for meta-learning.

We present *Metalic*, an in-context meta-learning approach to tackle the problem of protein fitness prediction in low-data settings, and make the following contributions:

- We introduce a method that efficiently combines in-context meta-learning with PLMs and fine-tuning for protein fitness prediction.
- We advance state-of-the-art (SOTA) for zero-shot protein fitness prediction on the ProteinGym benchmark (Notin et al., 2024).
- We attain strong performance for few-shot fitness prediction with 18 times fewer parameters.
- We ablate each component of our method to demonstrate the contributions of each part and underscore their necessity.
- We empirically validate the superiority of our method to alternative forms of meta-learning.

## 2 RELATED WORK

**Meta-Learning** Meta-learning aims to create a sample-efficient learning algorithm by training over a distribution of tasks. The goal is to learn algorithms such that they can rapidly adapt to new tasks during inference. This inference-time adaptation is often called the *inner loop*, for which there are two primary forms found in the literature: gradient-based meta-learning (Finn et al., 2017; Zintgraf et al., 2019) and in-context meta-learning (Mishra et al., 2017; Nguyen & Grover, 2022).

Gradient-based and in-context algorithms differ both in their computational efficiency and capacity for out-of-distribution generalization. Gradient-based approaches explicitly adapt model parameters within in the inner loop using standard gradient-based learning. Commonly, a parameter initialization is learned that can be adapted to new tasks with only a few gradient steps (Finn et al., 2017; Zintgraf et al., 2019; Vuorio et al., 2019). However, this comes with considerable computational overhead – due to meta-gradients when differentiating the inner loop – which makes gradient-based meta-learning less suitable for large models. Alternatively, in-context meta-learning adapts by conditioning a sequence model on a task-specific dataset in context. These methods condition on the data points over which gradient-based approaches would train (Santoro et al., 2016; Mishra et al., 2017; Beck et al., 2024; 2023b). Such methods are typically more sample- and compute-efficient than gradient-based methods, but perform worse on out-of-distribution tasks given the lack of explicit gradient-based learning in the inner loop (Beck et al., 2023a). Rather than training for in-context learning, the in-context learning of pre-trained large language models can also be used to perform meta-learning (Coda-Forno et al., 2023); however, this lacks generalization guarantees and requires fitting all tasks in context simultaneously, which is not possible given our data (Section 4.1).

In *Metalic*, we only train for in-context meta-learning, but find this is still compatible with task-specific fine-tuning. In the meta-reinforcement learning setting, this combination has been shown to be possible by increasing task-specific data for fine-tuning (Xiong et al., 2021). In contrast, we evaluate in the supervised setting and do not give increased data at inference time. While prior work has combined gradient-based and in-context meta-learning (Rusu et al., 2018; Vuorio et al., 2019), these works compute expensive meta-gradients to learn how to account for fine-tuning. Despite not explicitly meta-learning gradient-based adaptation, we find in-context meta-learning alone provides a strong foundation for subsequent fine-tuning and that both aspects are critical for high performance.

**Likelihood-Based Fitness Prediction with PLMs**   Leveraging pre-trained PLMs is standard practice in protein fitness prediction (Rives et al., 2021; Notin et al., 2023; Rao et al., 2021; Truong Jr & Bepler, 2024). In the few-shot setting, PLMs intended for sequence generation are repurposed by fine-tuning for protein fitness prediction (Rives et al., 2021). In the zero-shot setting, it is assumed that the fitness correlates with the likelihood of the proteins associated sequence of amino acids, as predicted by a PLM (Meier et al., 2021; Truong Jr & Bepler, 2024). Furthermore, if using a masked PLM, it is often assumed that each amino acid contributes independently to the fitness (Meier et al., 2021). In this work, we likewise leverage PLMs for protein fitness prediction. However, in contrast, we make use of additional data in the form of additional fitness prediction tasks on other proteins. Specifically, we meta-learn how to use a PLM for protein fitness prediction, rather than relying on assumptions. Only after meta-learning, do we fine-tune our model, as depicted in Fig. 1. Meta-learning across tasks lets us avoid restrictive model constraints and achieve SOTA performance. We will show that in-context meta-learning is necessary to achieve strong results in low-data settings.

**In-Context PLMs**   We build upon existing PLMs that make use of in-context data for protein fitness prediction (Notin et al., 2022; Truong Jr & Bepler, 2024; Notin et al., 2023; Rao et al., 2021). However, these methods do not *meta-learn* how to make use of their context. These methods either learn to use the context only for protein language modelling, and then assume that the likelihood from the generative model correlates with fitness (Truong Jr & Bepler, 2024; Notin et al., 2022; Rao et al., 2021), or they use the context for protein fitness prediction, but not by meta-learning over protein tasks (Notin et al., 2023). Of these ProteinNPT (Notin et al., 2023) is the most related to our method, since we use the same attention architecture to condition on fitness information about related proteins in-context, and it uses gradient steps to fine-tune to the target task. In comparison, our method meta-learns over many tasks how to make use of the fitness information, which we find to be critical (Section 4). Additionally, our method is the first to use the aforementioned procedure to allow fine-tuning and in-context conditioning on the very same context at inference time.

# 3   METHODS

## 3.1   PROBLEM SETTING

We consider a fitness prediction task, $\mathcal{T}$, to be defined by a dataset of the form $\mathcal{D}_{\mathcal{T}} = \{(x_i, y_i \equiv f_{\mathcal{T}}(x_i))\}_{i=1}^{N}$, where $x_i$ is a sequence of amino acids, and $y_i \in \mathbb{R}$ is the associated scalar fitness value assigned by the (unknown) underlying fitness function $f_{\mathcal{T}}$. In the few-shot setting, the task-specific data is typically split into non-overlapping support and query sets:

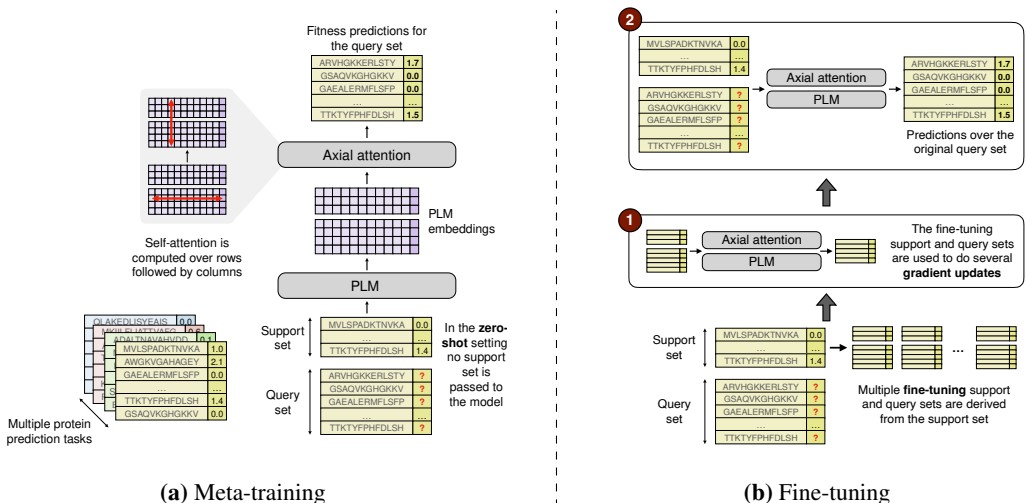

(a) Meta-training        (b) Fine-tuning

**Figure 2: An overview of _Metalic_.** Meta-training our model over many protein prediction tasks enables in-context learning (a). Fine-tuning the in-context learning on the support set requires sub-sampling smaller support and query sets, and enables generalization at test time (b).

$\mathcal{D}_{\mathcal{T}}^{(S)} = \{(x_i^{(S)}, y_i^{(S)})\}_{i=1}^{N^{(S)}}$ and $\mathcal{D}_{\mathcal{T}}^{(Q)} = \{(x_i^{(Q)}, y_i^{(Q)})\}_{i=1}^{N^{(Q)}}$ such that $\mathcal{D}_{\mathcal{T}}^{(S)} \cap \mathcal{D}_{\mathcal{T}}^{(Q)} = \emptyset$, with support and query set sizes, $N^{(S)}$ and $N^{(Q)}$, respectively. The support set provides data for task-specific adaptation, and the query set provides data for evaluating the adapted performance. The size of the support set is called the *shot*. Note, even with an empty support set (*zero-shot*), the model can still adapt to the task using information from the query set, i.e., related sequences without fitness scores.

Meta-learning for protein fitness prediction requires not just a single task, but multiple tasks, $\mathcal{D} = \mathcal{D}_1, \ldots \mathcal{D}_T$ over which to learn. The full dataset of tasks, $\mathcal{D}$, can be seen as defining a distribution of tasks which can be split into training and test tasks in the usual way. Concretely, for meta-learning in-context, the goal is to learn a function with parameters $\theta$ conditioned on the full support set and unlabelled inputs from query set, $f_\theta(\{x_i^{(Q)}\}_{i=1}^{N^{(Q)}}, \mathcal{D}_{\mathcal{T}}^{(S)})$. In our case, rather than directly predicting fitness values, we instead follow prior works that use a preference-based objective to rank the query set in order of fitness (Krause et al., 2022; Brookes et al., 2023; Hawkins-Hooker et al., 2024).

## 3.2 METALIC

**Architecture** Our work leverages the ProteinNPT architecture proposed by Notin et al. (2023) as an in-context PLM for fitness prediction. Here we briefly summarize the key elements, but defer the reader to Appendix A.2 and the original paper for full details. The architecture, along with the data we use for meta-leaning, are illustrated in Fig. 2a. A detailed illustration is give in Appendix A.6.

Protein sequences in both the support and query set are converted to per-residue embeddings using a pre-trained PLM; in our case we take the third layer of ESM2-8M (Lin et al., 2022). Fitness scores in the support set are projected to match the dimensionality of the residue embeddings using a linear layer, while the query set fitness embeddings share a single learned embedding. The protein sequence embeddings and fitness embeddings are then concatenated along the sequence dimension. Optionally, zero-shot fitness predictions from an auxiliary PLM can be embedded and concatenated in the same way, which we explore in Section 4.3. This tensor is then processed via axial attention blocks (Ho et al., 2019), each of which applies self-attention separately along and across the sequences. Axial attention reduces the computational complexity of self-attention from $O(K^2 L^2)$ to $O(K^2 + L^2)$, where $K$ is the shot and $L$ is the length of a protein. Finally, a multi-layer perceptron conditions on the fitness embedding and mean-pooled sequence embedding to predict each query value, i.e. $\{v_i^{(Q)}\}_{i=1}^{N^{(Q)}} = f_\theta(\{x_i^{(Q)}\}_{i=1}^{N^{(Q)}}, \mathcal{D}_{\mathcal{T}}^{(S)})$, which is then used to rank the proteins by fitness.

**Meta-Training** Following prior works that use a preference-based objective (Krause et al., 2022; Brookes et al., 2023; Hawkins-Hooker et al., 2024), we reframe the relative score prediction of two sequences as binary classification, predicting whether sequence $x_i^{(Q)}$ has a higher fitness than $x_j^{(Q)}$:

$$p\left(y_i^{(Q)} > y_j^{(Q)}\right) = \sigma\left(v_i^{(Q)} - v_j^{(Q)}\right), \tag{1}$$

where $\sigma$ is a sigmoid function and the dependency of the query values on $\theta$ has been dropped for brevity. This classifier is optimized with respect to every pairwise comparison between sequences in the query set corresponding to the loss function:

$$\mathcal{L}(\theta, \mathcal{D}_{\mathcal{T}}^{(\text{Q})}, \mathcal{D}_{\mathcal{T}}^{(\text{S})}) = -\sum_{i=1}^{N^{(\text{Q})}} \sum_{\substack{j=1 \\ j \neq i}}^{N^{(\text{Q})}} \mathbb{I}\left(y_i^{(\text{Q})} > y_j^{(\text{Q})}\right) \log \sigma\left(v_i^{(\text{Q})} - v_j^{(\text{Q})}\right), \tag{2}$$

where $\mathbb{I}$ is an indicator function. Intuitively, this is optimising $N^{(\text{Q})} \times (N^{(\text{Q})} - 1)$ binary classification problems. Note that we only compute the loss over the query set to avoid encouraging memorization of the support set. Adapting this to meta-learning (Fig. 2a), the objective becomes to find the parameterization that minimizes the loss across the task distribution,

$$\mathcal{J}(\theta, \mathcal{D}) = -\mathbb{E}_{\mathcal{D}_{\mathcal{T}} \in \mathcal{D}} \mathbb{E}_{(\mathcal{D}_{\mathcal{T}}^{(S)}, \mathcal{D}_{\mathcal{T}}^{(Q)}) \in \mathcal{D}_{\mathcal{T}}} \mathcal{L}(\theta, \mathcal{D}_{\mathcal{T}}^{(\text{Q})}, \mathcal{D}_{\mathcal{T}}^{(\text{S})}). \tag{3}$$

**Fine-Tuning** *Metalic* uses fine-tuning, during inference, in order to enable generalization, without having to account for fine-tuning during meta-training. This process is depicted in Fig. 2b.

The combination of in-context learning and fine-tuning creates a unique problem. Since fine-tuning occurs at inference time, labels for the query set, $\{y_i^{(\text{Q})}\}_{i=1}^{N^{(\text{Q})}}$, are not available for training. While labels for the support set, $\{y_i^{(\text{S})}\}_{i=1}^{N^{(\text{S})}}$, are available, propagating gradients from the support set would encourage memorization of the support set, since the labels are also passed as input in-context. While prior methods that combine in-context and gradient-based meta-learning (Rusu et al., 2018; Vuorio et al., 2019) would encounter this issue, this problem is exacerbated for *Metalic*. Whereas prior methods compress inputs to a representation with a constant number of dimensions, *Metalic* uses self-attention, which scales with the number of inputs, allowing them to be stored without compression. Moreover, whereas prior methods use meta-gradients that could adjust the gradient update procedure so as to be useful for generalization and not memorization, *Metalic* does not take into account the fine-tuning process during meta-training.

We address the issue of memorization by sub-sampling from the support set. The fine-tuning procedure is the same as during meta-training, with the exception that the support set is sub-sampled. In order to compute updates on a single support set, the support set is sub-sampled into multiple smaller support and query sets, $\mathcal{D}_{\mathcal{T}}^{(\text{S}')} \subseteq \mathcal{D}_{\mathcal{T}}^{(\text{S})}$ and $\mathcal{D}_{\mathcal{T}}^{(\text{Q}')} \subseteq \mathcal{D}_{\mathcal{T}}^{(\text{S})}$. Concretely, this corresponds to fine-tuning on unseen data using the objective:

$$\mathcal{J}(\theta, \mathcal{D}_{\mathcal{T}}^{(\text{S})}) = -\mathbb{E}_{(\mathcal{D}_{\mathcal{T}}^{(S')}, \mathcal{D}_{\mathcal{T}}^{(Q')}) \in \mathcal{D}_{\mathcal{T}}^{(\text{S})}} \mathcal{L}(\theta, \mathcal{D}_{\mathcal{T}}^{(\text{Q}')}, \mathcal{D}_{\mathcal{T}}^{(\text{S}')}). \tag{4}$$

After fine-tuning, *Metalic* conditions on the complete support set, allowing no data to go to waste.

Using *Metalic*'s unique combination of in-context meta-learning followed by fine-tuning, we enable the generalization of extensive fine-tuning, while also precluding expensive computation. If a typical gradient-based meta-learning method requires $O(m)$ meta-gradients and $O(mn)$ regular gradients for meta-training, *Metalic* requires no meta-gradients and $O(m)$ regular gradients, constituting a linear reduction with superior performance to efficient alternatives, as demonstrated in Section 4.6.

## 4 EXPERIMENTS

In this section, we evaluate *Metalic* on fitness prediction tasks from the ProteinGym benchmark (Notin et al., 2024). We evaluate in the zero-shot setting with no support data, and the few-shot setting with limited support data. To establish SOTA results in the zero-shot setting, we first compare to the predictions provided by ProteinGym for the baseline models. To establish strong performance in the few-shot setting, since predictions are not provided, we train baselines from Hawkins-Hooker et al. (2024). While *Metalic* does not achieve SOTA results in evaluations on proteins that have multiple mutations (multi-mutant proteins), we demonstrate that the performance grows as we add more meta-training tasks, providing a path forward for *Metalic* in the multi-mutant setting in the future. We perform ablations of *Metalic*, to show the benefits of meta-learning, in-context learning, and fine-tuning. Finally, we compare to the gradient-based method, Reptile (Nichol et al., 2018), to show that taking account of gradients during training is an unnecessary computational burden.

## 4.1 EXPERIMENTAL SETUP

In our experiments, we focus on ProteinGym deep mutational scans. Each task in ProteinGym each measures one property on a set of proteins that all differ by one amino acid, or multiple amino acids, from a reference wild-type protein. We have 121 single-mutant tasks and 68 multi-mutant tasks from ProteinGym. From these, we evaluate over eight held-out single-mutant tasks, and five held-out multi-mutant tasks, following Notin et al. (2023); Hawkins-Hooker et al. (2024). This leaves 113 single-mutant and 68 multi-mutant tasks for training when evaluating single mutants (Sections 4.2 and 4.3), and 121 single-mutant and 63 multi-mutant tasks for training when evaluating multiple mutants (Section 4.4). All fitness values are standardized by subtracting the mean and dividing by the standard deviation by task. Note that there are additional tasks in ProteinGym we do not consider. Specifically, we do not consider multi-mutant tasks that have overlapping proteins with single-mutant tasks, to make evaluation more difficult. We also ignore additional tasks in which the maximum protein length is > 750, to fit the backward pass on an Nvidia A100-80Gb device.

We use a query set size of $N^{(Q)} = 100$, and the size of the support set is determined by our evaluation setting and is one of three sizes: $N^{(S)} = 0$, 16, or 128. We also use an additional set of 128 points just for early stopping of the fine-tuning process, for all models, following the implementation of Hawkins-Hooker et al. (2024). We then evaluate remaining points in the task, with a maximum of 2,000 points total, by dividing the data into multiple query sets. If the model can fit a larger query size, as in non-meta-learning baselines, then we pass the remaining data as a single query set. If the data is not divisible by the query set size, it is left out from evaluation. However, we sample the support data over three independent samples, avoiding systemic exclusion.

All evaluation uses the Spearman rank correlation, in line with prior work (Notin et al., 2023; Truong Jr & Bepler, 2024; Hawkins-Hooker et al., 2024). We compute the Spearman correlation per task, and then average over tasks. For all evaluations of our models, we compare over three seeds for training and report the mean and standard deviation. Each context, consisting of a support and query set, consists of ≤ 171,000 tokens. We meta-train for 50,000 updates and fine-tune for 100. Fine-tuning uses the same procedure, including an Adam optimizer and cosine learning rate scheduler, but fine-tuning skips the learning rate warm-up, so the scheduling has little effect. Using a single Nvidia A100-80Gb, training our model takes roughly 2 to 8 days per seed, depending on support size and frequency of fine-tuned evaluation.

## 4.2 ZERO-SHOT

The first setting we evaluate is the zero-shot performance of our model, with no support set for fine-tuning. In Table 1 we compare against predictions provided by ProteinGym for each baseline to compute the zero-shot Spearman correlation ($\rho$). We compare to provided predictions, on our data splits, to enable a fair comparison to the strongest models available without retraining each baseline from scratch ourselves. We include the best performing model, and notable models, as baselines. We find that *Metalic* outperforms every reported baseline and is SOTA at zero-shot prediction.

**Table 1: Spearman correlation in the zero-shot setting.** Results are computed using predictions provided by ProteinGym. Using a single PLM, in the zero-shot setting, renders the baselines deterministic. For comparison, we report our best model, and the mean and standard deviation, for quantifying the variation in meta-training. *Metalic* achieves SOTA performance in either case.

| Model Name | Spearman Correlation |
|---|---|
| *Metalic* (max) | **.484** |
| *Metalic* (mean) | **.482** ± .002 |
| VESPA | .464 |
| TranceptEVE-Medium | .457 |
| ESM1-v-650M | .437 |
| Tranception-Medium | .427 |
| Progen2-Medium | .419 |
| ESM2-650M | .399 |
| MSA Transformer | .398 |
| ESM-IF1 | .365 |
| ESM2-8M | .121 |

Our method significantly outperforms strong baselines with many more parameters, such as ESM1-v-650M. The 8 million parameter PLM used by our method, ESM2-8M, without *Metalic*, achieves a score of only .121 $\rho$, demonstrating the large contribution of our meta-learning procedure. The next strongest method after ours is VESPA (Marquet et al., 2022). VESPA optimizes PLM embeddings to predict a distinct set of binary annotations from 9,594 proteins, and uses these features to predict protein fitness by comparing to a reference wild-type embedding. Note, unlike our method, VESPA relies on strong assumptions to generalize and is specific to the zero-shot setting.

The strong performance of our method in the zero-shot setting can be attributed to meta-learning. Since there is no data for fine-tuning, the zero-shot performance increase over ESM2-8M derives entirely from our meta-training procedure. In this case, other methods generally assume that the PLM likelihood of a mutation correlates with fitness, whereas our model learns to make use of the information contained in PLM embeddings to make predictions zero-shot. Moreover, our method still conditions on an unlabeled query set, and the protein embeddings in that query set, which allow for meta-learning a form of in-context unsupervised adaptation.

## 4.3 FINE-TUNING RESULTS

In Table 2 we report Spearman correlation with a support set of size $N^{(\text{S})} = 0$, 16, and 128, and we compare to baselines that we train and evaluate ourselves over three random seeds. We re-train these methods using the models, following Hawkins-Hooker et al. (2024), to provide a comparison over multiple seeds between these methods in a range of practical low-data settings. All models use the same preference-based loss function as *Metalic*, for a fair comparison, and none use ensembling.

We also train *Metalic*-AuxIF, which allows *Metalic* to condition on zero-shot predictions from an additional PLM, as introduced in Section 3.2. We choose ESM-IF1 (Hsu et al., 2022) as the auxiliary input, since it contains embeddings derived from inverse folding that summarize protein structure. Note that *Metalic*-IF1 uses 170 million parameters, whereas *Metalic* uses 36 million (including the ESM2-8M embedding), so the auxiliary predictions significantly increases the total parameters.

Again, we find that *Metalic* has the strongest performance in the 0-shot and 16-shot settings, and has comparably strong performance in the 128-shot setting, with 18 times fewer parameters. Moreover, *Metalic* also outperforms contemporary models, such as PoET (Truong Jr & Bepler, 2024), that make use of additional evolutionary information, in the form of multi-sequence alignment, and in-context learning. Note that ESM2-8M and ProteinNPT are the worst performing methods in the few-shot setting. This results suggests that the effectiveness of *Metalic* does not come from the ESM2-8M embedding, nor the ProteinNPT architecture, but rather from the meta-learning itself.

Additionally, we see that *Metalic*-AuxIF, improves results. This result is significant because ESM2-8M and ESM-IF1, the two PLMs used by *Metalic*-AuxIF, are the worst performing methods in the zero-shot setting (Table 1), with ESM2-8M also weak in the few-shot setting. Thus, the effectiveness of *Metalic*-AuxIF comes entirely from meta-learning: We can learn how and when to rely on features from each of these weak predictors of protein fitness, to combine them into a strong predictor.

Consistent with the motivation of meta-learning, results are strongest when the data is most limited. Meta-learning adds an additional training stage to learn prior beliefs and inductive biases from related data. The more limited, the more relying on prior data is useful.

**Table 2: Spearman correlation for the 0, 16, and 128-shot setting.** Baseline results are recomputed. Standard deviation is provided over three seeds. *Metalic* matches or exceeds all baselines.

| Model Name | $n = 0$ | $n = 16$ | $n = 128$ |
|---|---|---|---|
| *Metalic* | **.482** $\pm$ .002 | **.484** $\pm$ .001 | .552 $\pm$ .009 |
| *Metalic*-AuxIF | **.498** $\pm$ .008 | **.500** $\pm$ .002 | .556 $\pm$ .005 |
| ESM1-v-650M | .384 $\pm$ .000 | .452 $\pm$ .000 | .553 $\pm$ .000 |
| ESM2-8M | .105 $\pm$ .000 | .226 $\pm$ .000 | .406 $\pm$ .000 |
| PoET | .416 $\pm$ .003 | .475 $\pm$ .026 | **.588** $\pm$ .006 |
| ProteinNPT | N/A | .192 $\pm$ .003 | .443 $\pm$ .003 |

**Table 3a: Multi-mutant Spearman correlation in the zero-shot setting.** Results are computed using predictions provided by ProteinGym on tasks with multiple mutations. *Metalic* is competitive, but not better than all baselines, due to lack of sufficient datasets with multiple mutants.

| Model Name | Multiples 0-Shot |
|---|---|
| *Metalic* (max) | .450 |
| *Metalic* (mean) | .436 ± .011 |
| *Metalic*-AuxIF (max) | .548 |
| *Metalic*-AuxIF (mean) | .533 ± .012 |
| ESM-IF1 | **.590** |
| TranceptEVE-Medium | .529 |
| Tranception-Medium | .513 |
| MSA Transformer | .503 |
| VESPA | .408 |
| ESM2-650M | .345 |
| Progen2-Medium | .305 |
| ESM2-8M | .289 |
| ESM1-v-650M | .279 |

**Figure 3b: Multi-mutant Spearman Correlation by Shot.** Spearman correlation for the 0, 16, and 128-shot, over three seeds, sorted by zero-shot performance. For complete results, including for *Metalic*-AuxIF, see Appendix A.5.

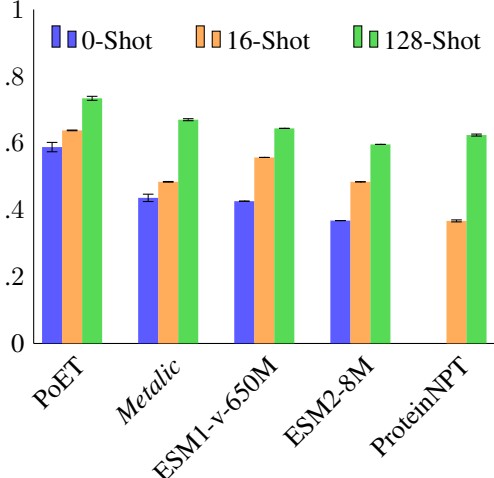

## 4.4 MULTIPLE MUTANTS

In this section we evaluate *Metalic* on tasks where proteins have multiple mutations. In Table 3a, we see that *Metalic* has strong performance, but does not outperform all baselines. However, with auxiliary inverse folding predictions (*Metalic*-AuxIF), *Metalic* is only outperformed by ESM-IF1 itself. These results indicate that ESM-IF1 is particularly strong on the eight held-out evaluation tasks, and that we can recover most of its performance by incorporating its predictions into our model. Note that while ESM-IF1 is strong in the multi-mutant setting, it is among the worst performing models in the single-mutant setting. In contrast, our method is strong in both settings, since it can learn to leverage ESM-IF1 predictions when they are helpful, and ignore them when they are not.

By comparing across all shot settings, we can see that *Metalic* is competitive in the multi-mutant setting (Fig. 3b), but no longer SOTA, which we hypothesize is due to limited multi-mutant training data. In Table 4a and Fig. 4b, we explore this hypothesis. Specifically, we see the performance of *Metalic* increases as the amount of training data, measured in tasks, increases. This trend holds true even when adding single mutant tasks, which are significantly different from the testing data. Providing more data is a path forward for strengthening the multi-mutant results.

**Table 4a: Multi-mutant Spearman correlation by training data.** Results are computed on eight tasks with multiple mutations with different amounts of single- and multi-mutant training tasks. We see performance of *Metalic* increases with more data, giving a path to improve results with future data collection. This trends holds even when we add single-mutant data, which is significantly different from the multi-mutant test data.

| Single-Mutant | Multi-Mutant | Multiples $n = 0$ |
|---|---|---|
| 121 | 63 | **.436 ± .011** |
| 121 | 30 | .417 ± .021 |
| 121 | 1 | .383 ± .010 |
| 0 | 63 | .376 ± .018 |
| 0 | 30 | .307 ± .022 |
| 0 | 1 | .047 ± .056 |

**Figure 4b: Multi-mutant spearman correlation by number of tasks.** *Metalic* with and without 121 additional single-mutant training tasks. (With 121 single-mutant is the default.) The performance of *Metalic* increases with more data, giving a path to improve results.

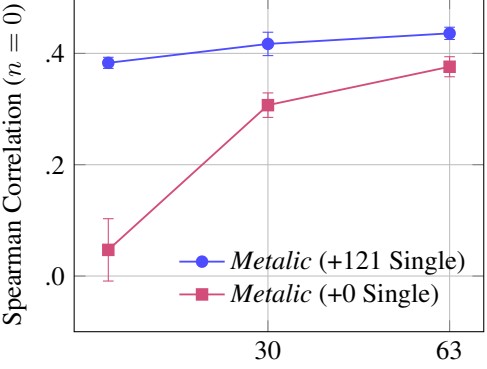

## 4.5 ABLATIONS

Here we report ablations of *Metalic*, evaluating on single-mutants (Table 5). Spearman correlation is reported in the zero-shot and 128-shot settings. The results justify all components of our method.

Most detrimental to performance is removing meta-training from *Metalic* (NoMetaTrain). This ablation is identical to *Metalic*, with the exception that there is no additional meta-learning stage over multiple protein landscapes. We see that without meta-learning, the initial zero-shot predictions have near zero correlation with the fitness and that the 128-shot predictions are critically impaired.

We also ablate the ability to attend to the rest of the proteins in context by turning off the column attention in the axial attention layers (NoICL), we ablate the fine-tuning stage of training (NoFT), and we ablate the preference loss by using mean squared error instead (No-Pref). Note that when we remove the fine-tuning, and have a non-zero support size, we allow the early stopping data to be passed in-context, to not unfairly advantage fine-tuning with additional data. Ablating in-context learn-

**Table 5: Ablations in the 0 and 128-shot setting.** Results show the importance of fine-tuning, in-context learning, meta-training, and additional training tasks as an augmentation.

| Model Name | $n = 0$ | $n = 128$ |
|---|---|---|
| *Metalic* | $\mathbf{.482} \pm .002$ | $\mathbf{.552} \pm .009$ |
| *Metalic*-NoFT | $\mathbf{.482} \pm .002$ | $.488 \pm .012$ |
| *Metalic*-NoPref | $.465 \pm .011$ | $.520 \pm .006$ |
| *Metalic*-NoICL | $.441 \pm .002$ | $.529 \pm .002$ |
| *Metalic*-NoMetaTrain | $-.046 \pm .018$ | $.346 \pm .003$ |

ing decreases performance in both settings, indicating an ability to adapt in an unsupervised fashion even from the query set alone. Ablating fine-tuning decreased performance in the 128-shot setting, indicating the need for fine-tuning for generalization to held-out data. Ablating the preference-based loss decreased performance in both settings, in line with recent literature (Krause et al., 2022; Brookes et al., 2023; Hawkins-Hooker et al., 2024). An additional ablation is given in Appendix A.3.

## 4.6 GRADIENT-BASED META-LEARNING

In this section, we compare *Metalic* to the same architecture but trained with Reptile (Nichol et al., 2018), an efficient method for gradient-based meta-learning. Unlike *Metalic*, Reptile does not use in-context learning and modifies the outer-loop during meta-training to take into account the subsequent fine-tuning. While accounting for the fine-tuning during meta-training comes with increased computational costs and can increase bias and variance (Vuorio et al., 2021), Reptile provides a simplified algorithm that can be run more efficiently. The primary differences between Reptile and *Metalic* are that Reptile adjusts the meta-learning loss to account for fine-tuning during meta-training, while *Metalic* performs in-context learning. Reptile details are provided in Appendix A.1.

Although Reptile is more compute-efficient than other gradient-based methods, it still performs gradient updates in the inner loop during meta-training, making it computationally expensive relative to *Metalic*. Our method uses 100 updates on the support data for fine-tuning. Reptile uses inner-loop gradient updates during both meta-training and fine-tuning. Due to compute limitations, we cannot use 100 steps for each forward pass of meta-training. Thus, we evaluate Reptile with 3 inner-loop gradient steps during meta-training and 3 during fine-tuning, so that train and test match (Reptile-3-3), and we evaluate Reptile with 3 inner-loop gradient steps during meta-training and 100 during fine-tuning, so that test time matches our method (Reptile-3-100). Note that even Reptile-3-3 uses three times more compute than *Metalic*. Finally, we evaluate Reptile-3-100 with *Metalic*, to see whether they can be used in conjunction (*Metalic*-Reptile). We train for 50,000 steps, and also report *Metalic* after 150,000 steps, to match the number of gradient computations as the Reptile models.

**Table 6: Comparison to Reptile.** Reptile (Nichol et al., 2018) is a gradient-based meta-learning method. Results are reported after 50,000 steps, with *Metalic* additionally reported after 150,000 steps, to allow for an equal number of gradient computations as the methods using Reptile. Results show that accounting for fine-tuning during meta-training using Reptile is unnecessary.

| Model Name | Meta-Training Steps | Total Gradient Computations | $n = 128$ |
|---|---|---|---|
| *Metalic* (150k) | 150,000 | 150,000 | $\mathbf{.562} \pm .004$ |
| *Metalic*-Reptile | 50,000 | 150,000 | $\mathbf{.562} \pm .004$ |
| *Metalic* (50k) | 50,000 | 50,000 | $.552 \pm .009$ |
| Reptile-3-100 | 50,000 | 150,000 | $.539 \pm .001$ |
| Reptile-3-3 | 50,000 | 150,000 | $.499 \pm .008$ |

Results are reported in Table 6. We evaluate in the 128-shot setting. Note that Reptile is not applicable in the zero-shot setting, as it requires some data for fine-tuning. We observe that without controlling for the amount of computation, *Metalic* (50k) outperforms both Reptile variants. We also observe that *Metalic*, in conjunction with Reptile (*Metalic*-Reptile), outperforms Reptile and *Metalic* independently. However, the improvement over *Metalic* is marginal, and the increased computation (by a factor of 3) is large. Controlling for the total number of gradient computations, *Metalic* (150k) achieves a Spearman correlation comparable to the combined *Metalic*-Reptile. Thus, Reptile performs poorly in isolation, and, while Reptile can be used with *Metalic*, the combination increases implementation complexity with no clear advantage. This result is in line with previous work showing that accounting for gradient updates during meta-training (in their case, without in-context learning) can be detrimental when there are a limited number of tasks for meta-training or a limited amount of data for the inner-loop (Gao & Sener, 2020; Triantafillou et al., 2020). Collectively, this evidence further justifies fine-tuning only after meta-training.

## 5 ANALYSIS

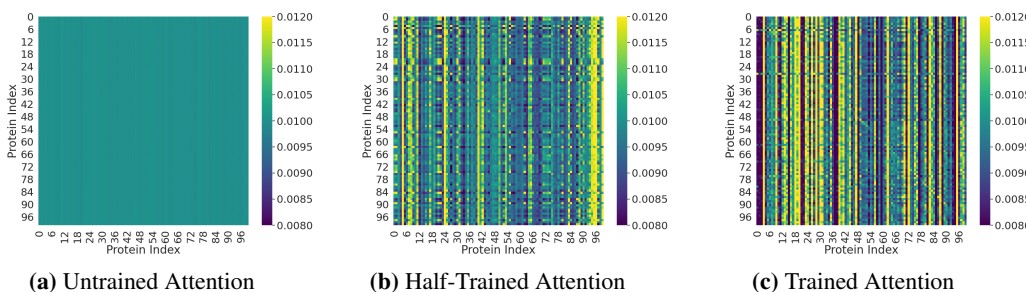

**(a)** Untrained Attention  **(b)** Half-Trained Attention  **(c)** Trained Attention

**Figure 5: Attention maps.** We present axial attention maps over the query set in the zero-shot setting. Each row shows attention to other proteins in context, normalized to one by row, averaged over layers and mutation location. Attention at step 1 (a) 25,0000 (b) and 50,000 (c). Each protein learns to pay attention to itself, while still attending to other significant proteins in context.

Here, we briefly investigate the in-context learning abilities of *Metalic* via attention maps. In Section 4, we show that in-context learning is vital to *Metalic* by ablating the attention between proteins and showing decreased performance. Notably, the in-context learning was beneficial not only in the few-shot setting, but also in the zero-shot setting. This suggests an interesting phenomenon: The emergence of unsupervised in-context learning from the query set alone. To confirm this phenomenon, in Fig. 5 we show the attention maps in the axial attention layers between proteins in the query set. Over the course of training, we observe the emergence of bright vertical lines, indicating some significantly informative proteins to which all others proteins attend. Moreover, we observe no rows that are entirely dark off the diagonal entries. Thus, no protein attends only to itself. Together, these results further corroborate the necessity of in-context meta-learning.

## 6 CONCLUSION

In this paper, we have demonstrated state-of-the-art results on a standard protein fitness prediction benchmark in low-data settings. To do so, we proposed *Metalic*, which makes use of both in-context meta-learning and subsequent fine-tuning. Critically, we have demonstrated the ability of meta-learning to take advantage of additional data from other proteins fitness prediction tasks, while remaining computationally tractable by deferring fine-tuning to test time alone. Unique within the meta-learning literature, we show that in-context meta-learning provides a useful initialization for further fine-tuning, and can make use of test time data for both fine-tuning and in-context learning. *Metalic* additionally demonstrates the ability to learn from the query set alone (zero-shot), performing unsupervised in-context learning. Future work could investigate leveraging *Metalic*'s unique in-context learning ability to act as an auto-regressive fitness model (i.e., a world model in the meta-reinforcement learning setting (Beck et al., 2023a)), for optimally trading off exploration and exploitation when designing novel proteins. Given its efficacy at leveraging additional data, we believe that meta-learning in-context will play a crucial role in advancing protein fitness prediction, with *Metalic* being a foundational step in that direction.

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

## A APPENDIX

### A.1 REPTILE DETAILS

This section provides additional details on how Reptile (Nichol et al., 2018) works. In order to account for fine-tuning during meta-training, gradient-based methods generally compute a *meta-gradient* that requires the computation of higher order derivatives, which can be computationally intractable for a large model. The costs can be especially burdensome when many gradient steps are needed for out of distribution adaptation. For this reason, Reptile avoids meta-gradients by changing optimization during meta-training. Here, the new parameters are updated, not by gradient descent, but rather by moving toward the mean, after each theta is adapted to a task, $\theta_{\mathcal{T}}$, by fine-tuning on that task. From time-step $t$ to $t + 1$ of this outer-loop optimization process, Reptile can be written:

$$\theta^{t+1} = \theta^t + \beta\mathbb{E}_{\mathcal{D}_{\mathcal{T}}\in\mathcal{D}}[\theta_{\mathcal{T}}^t - \theta^t]. \tag{5}$$

We had to choose several implementation details for Reptile. First, note that Reptile sub-samples the support set during fine-tuning and has no distinct query set. In our implementation, we sub-sample mini-batches of size 50 to match the query size of *Metalic* for sub-sampling in the 128-shot setting. Reptile also has several options for the outer loop optimization. We choose to use the batched version with a batch size of 4 to match *Metalic*. Rather than using the direct update given in Equation (5), we take the difference over the learning rate, $(\theta_{\mathcal{T}}^t - \theta^t)/\alpha$ as an approximation of a gradient to be used with the Adam optimizer, as suggested by Nichol et al. (2018) and validated in Appendix A.2.

### A.2 HYPER-PARAMETERS

Since we build upon the axial attention of ProteinNPT (Notin et al., 2023), we follow their choice for most hyper-parameters, with a few exceptions. Most notably, in our experiments, we use the third layer of ESM2 as an embedding for each protein, given the strong performance and reduced number of parameters. Additionally, we use a ranking loss from Hawkins-Hooker et al. (2024), and do not use the CNN or additional inputs (such as zero-shot predictions) from ProteinNPT. We likewise found conditioning on the wild-type unhelpful. The same set of hyper-parameters are used for each setting: 0-shot, 16-shot, and 128-shot, and for multi-mutant results. We used the same learning rate for fine-tuning *Metalic* as for meta-training. We found the default ProteinNPT learning rate to be too large, and decreasing by a factor of 5 to be sufficient. Consistent with our baselines, we select hyper-parameters using the single-mutant results and then use the same hyper-parameters for the multi-mutant setting, following Hawkins-Hooker et al. (2024); Notin et al. (2023). Complete details on hyper-parameters used are available in Table 7. We tuned relatively few of the hyper-parameters of our method, and mostly tuned over a single seed. There is likely room for improvement in the hyper-parameter selection of *Metalic*.

For the majority of baselines no tuning was required, other than Reptile. For the baselines, we used reference predictions for Table 1 and reference implementations for Table 2, neither of which required tuning. For Reptile, the update in the outer-loop can be uses as written in Equation (5), or $(\theta_{\mathcal{T}}^t - \theta^t)/\alpha$ can be interpreted as a gradient for use with the Adam optimizer Nichol et al. (2018). We experiment with both and find the use of the Adam optimizer to be superior in performance. All results in the main body use the Adam optimizer for Reptile. In order to evaluate the method without the Adam optimizer, we re-tune $\beta$ in Equation (5), which is the outer-loop learning rate. We leave the inner-loop learning rate, $\alpha$, fixed at the learning rate for *Metalic*, $3e^{-5}$, which uses the same learning rate for the inner- and outer-loop. Given the increased computational cost of Reptile, we use a single seed over three learning rates for 10,000 steps. We tune over the following learning rates ($\beta$): the learning rate for *Metalic*, which is $3e^{-5}$; a learning rate of 1, which corresponds to the learning rate of *Metalic* if you interpret $(\theta_{\mathcal{T}}^t - \theta^t)/\alpha$ as the gradient in a gradient-based update (Nichol et al., 2018); and a learning rate of $\frac{1}{n}$, where $n$ is the number of inner-loop updates (3 in this case), which corresponds to the learning rate of *Metalic* if you interpret $(\theta_{\mathcal{T}}^t - \theta^t)/(\alpha n)$ as the gradient in a gradient-based update[1]. We found a learning rate of 1 to perform best, in line with the

---

[1]The standard gradient update, $\theta^{t+1} = \theta^t + \alpha\nabla$, with $\nabla = (\theta_{\mathcal{T}}^t - \theta^t)/(\alpha n)$, gives $\theta^{t+1} = \theta^t + \alpha(\theta_{\mathcal{T}}^t - \theta^t)/(\alpha n)$, which is the same update as Equation (5) with $\beta = \frac{1}{n}$, omitting the expectation for brevity.

**Table 7:** Hyper-Parameters for *Metalic*. Where different, an earlier set of hyper-parameters used for ablations and Reptile comparisons are given in parentheses.

| Hyper-Parameter | Description | Value |
|---|---|---|
| Training Steps | The total number of training steps in meta-training | 50,000 |
| Warm-Up Steps | The number of training steps spent linearly warming up, preceding cosine decay | 5,000 |
| Batch Size | The number of contexts evaluated per training step. Note that gradient accumulation is used for each context in the batch, so this scales training time linearly. | 4 |
| Weight Decay | Weight decay applied to non-bias parameters only | 5e-3 |
| Learning Rate | The learning rate for meta-training and fine-tuning | 6e-5 |
| Min LR Fraction | The minimum fraction of the LR maintained during the cosine decay in learning rate scheduling | 1e-5 |
| Adam Eps | The epsilon value for the Adam optimizer | 1e-8 |
| Adam Beta1 | The beta1 value for the Adam optimizer | 0.9 |
| Adam Beta2 | The beta2 value for the Adam optimizer | 0.999 |
| Gradient Clip Value | The maximum norm allowed for the gradient | 1.0 |
| ESM embed model | The full name for the ESM2 model used | esm_t6_8M_UR50D |
| ESM embed layer | The layer from the ESM2 model used as an embedding | 3 |
| Number Fine-tune Steps | The number of gradient updates for fine-tuning after meta-training | 100 |
| Num ProteinNPT Layers | The number of layers using axial attention, as in ProteinNPT | 5 |
| Condition on Pooled Sequence | Whether each sequence is pooled or ignored after axial attention | True |
| MLP Layer Sizes | The number and size of fully connected layers after axial attention | [768,768,768,768] |
| Embed Dim | The embedding dimension for all inputs including the protein sequences and fitness values | 768 |
| Axial Forward Embed Dim | The hidden size of the feed-forward layer within the ProteinNPT layer | 400 |
| Attention Heads | The number of heads in self-attention | 4 |
| Dropout Prob | The probability of dropout during training and fine-tuning for layers other than axial attention | 0.0 |
| Attention Dropout | The probability of dropout during training and fine-tuning for axial attention layers | 0.1 |
| Num Single Tasks | The total number of single-mutant tasks available. These tasks are included for meta-training even when testing on multi-mutants. Eight are held-out for evaluation when evaluating single-mutant performance. | 121 |
| Num Multiple Tasks | The total number of multi-mutant tasks available. These tasks are included for meta-training even when testing on single-mutants. Five are held-out for evaluation when evaluating multi-mutant performance. | 68 |
| Warm-up During Fine-tuning | Whether or not to use the linear warm-up from the learning rate scheduler during fine-tuning. | False |

outer loop learning rate of *Metalic* and the interpretation of the gradient from Nichol et al. (2018). For Adam, there is no $\beta$, and we do not re-tune Adam's outer-loop learning rate. When using Adam, the gradient is interpreted as $(\theta_{\mathcal{T}}^t - \theta^t)/\alpha$. The results of tuning the learning rate without Adam, suggests a learning rate of $3e^{-5}$ with this gradient interpretation. Indeed, we confirm that when using Adam with this learning rate, and this gradient interpretation, it works better than using the alternative Reptile update rule with any learning rate. Results of the optimizer and learning rate tuning are presented in table Table 8. Note that these results used an earlier set of hyper-parameters.

**Table 8:** Reptile tuning results for the 128-shot setting. Results are trained over 10,000 steps for one seed each. The Adam optimizer performs the best. Both Adam, with an outer-loop learning rate of $3e^{-5}$, and Equation (5), with $\beta = 1$, correspond to the same scale in the outer loop.

| Model Name | Adam ($3e^{-5}$) | $\beta = 1.$ | $\beta = .333$ | $\beta = 3e^{-5}$ |
|---|---|---|---|---|
| Reptile-3-3 | **.452** | .420 | .350 | .091 |
| Reptile-3-100 | **.472** | .444 | .396 | .181 |
| *Metalic*-Reptile | **.483** | .476 | .411 | .190 |

### A.3 AUGMENTATION ABLATION

In this section we provide an additional ablation. Specifically, we ablate the augmentation of single-mutant training data with multi-mutant data (NoAug). The result is reported in Table 9. Ablating the multi-mutant augmentation decreased performance in the zero-shot setting, with comparable 128-shot performance, indicating the benefit of additional meta-training data when fine-tuning data is most limited.

**Table 9: Augmentation Ablation in the 0 and 128-shot setting.** Results show the importance of augmenting with multi-mutant data in the zero-shot setting.

| Model Name | $n = 0$ | $n = 128$ |
|---|---|---|
| *Metalic* | **.482** $\pm$ .002 | **.552** $\pm$ .009 |
| *Metalic*-NoAug | .464 $\pm$ .011 | **.558** $\pm$ .002 |

### A.4 FINE-TUNING WARM-UP

*Metalic* does not use a warm-up period for fine-tuning as it does for meta-training. This decision was made since the warm-up normally occurs for 5,000 steps, but fine-tuning only occurs for 100 steps. As a compromise, we evaluate a warm-up period for fine-tuning, after taking a single gradient step with the full learning rate (*Metalic*-FTWarmUp). We still see that not using a warm-up for fine-tuning is superior. Results can be see in Table 10.

**Table 10:** Spearman correlation with and without warm-up.

| Model Name | $n = 128$ |
|---|---|
| *Metalic* | **.552** $\pm$ .009 |
| *Metalic*-FTWarmUp | .539 $\pm$ .007 |

## A.5 MORE MULTI-MUTANT RESULTS

In this section we present a table of additional results for tasks with multiple mutants, including the *Metalic*-AuxIF models.

In Table 12, we see the performance of *Metalic* increases as the amount of training data, measured in tasks, increases, providing more data is a path forward for strengthening the multi-mutant results. Here, we also see that the trend changes for *Metalic*-AuxIF, since it performs better with more data, but only if that data is multi-mutant data. This makes sense, since the auxiliary ESM-If1 predictions are most effective for multi-mutants, and training on single-mutants could discourage relying on them.

**Table 11:** Spearman correlation for the 0, 16, and 128-shot setting for multi-mutant tasks.

| Model Name | $n = 0$ | $n = 16$ | $n = 128$ |
|---|---|---|---|
| *Metalic* (Both) | $.436 \pm .011$ | $.484 \pm .001$ | $.670 \pm .003$ |
| *Metalic*-AuxIF (Both) | $.480 \pm .007$ | $.500 \pm .002$ | $.693 \pm .006$ |
| *Metalic*-AuxIF (MultiTrain) | $.533 \pm .012$ | $.577 \pm .008$ | $.692 \pm .003$ |
| ESM1-v-650M | $.426 \pm .000$ | $.557 \pm .000$ | $.644 \pm .000$ |
| ESM2-8M | $.368 \pm .000$ | $.484 \pm .000$ | $.596 \pm .000$ |
| PoET | $\mathbf{.588} \pm \mathbf{.014}$ | $\mathbf{.638} \pm \mathbf{.001}$ | $\mathbf{.734} \pm \mathbf{.006}$ |
| ProteinNPT | N/A | $.367 \pm .003$ | $.624 \pm .003$ |

**Table 12: Multi-mutant Spearman correlation by training data.** Results are computed using predictions provided by ProteinGym on five tasks with multiple mutations with different amounts of single- and multi-mutant training tasks. We see performance of *Metalic* increases with more data, giving a path for future data collection. We also see here that *Metalic*-AuxIF performs better when trained with only multi-mutant data. This makes sense, since the augmented predictions are most effective for multi-mutants, and training on single-mutants could discourage relying on them. An asterisk (*) represents the default training data regime for that method.

| | | | Multiples |
|---|---|---|---|
| Model Name | Single-Mutant Tasks | Multi-Mutant Tasks | $n = 0$ |
| *Metalic* (Both)* | 121 | 63 | $\mathbf{.436} \pm \mathbf{.011}$ |
| *Metalic* (MultiTrain) | 0 | 63 | $.376 \pm .018$ |
| *Metalic* (MultiTrainHalf) | 0 | 30 | $.307 \pm .022$ |
| *Metalic*-AuxIF (Both) | 121 | 63 | $.480 \pm .007$ |
| *Metalic*-AuxIF (MultiTrain)* | 0 | 63 | $\mathbf{.533} \pm \mathbf{.012}$ |
| *Metalic*-AuxIF (MultiTrainHalf) | 0 | 30 | $.517 \pm .021$ |

## A.6 DETAILED ARCHITECTURE

Here, we present Fig. 6, which gives a more detailed version of the architecture diagram.

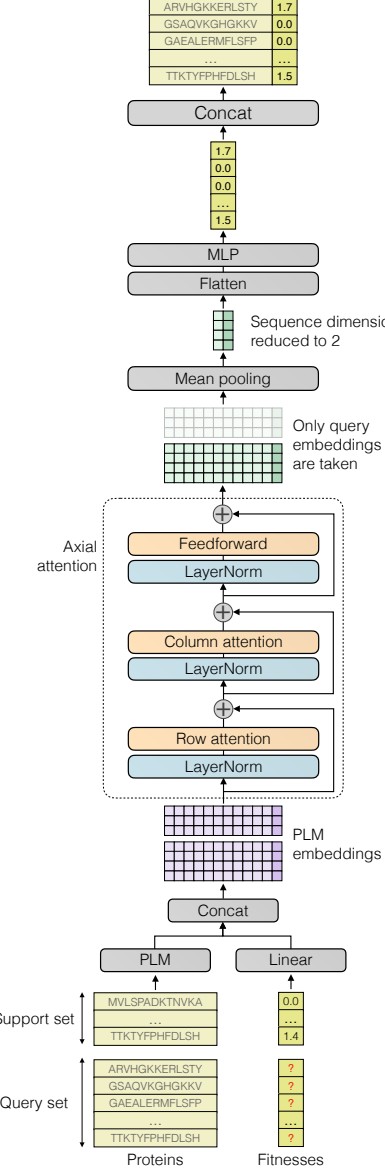

**Figure 6: Detailed diagram of architecture.** Note that fitness embeddings are computed using a learned linear projection for the support set and a shared learned embedding vector for the query set.

