# OpenReview forum: "Metalic: Meta-Learning In-Context with Protein Language Models"
_ICLR.cc/2025/Conference — ICLR 2025 Poster_

### Official Review · Reviewer_ZGWy · 2024-10-22

**Soundness:** 4
**Presentation:** 4
**Contribution:** 2
**Rating:** 5
**Confidence:** 4

**Summary:**

The authors present Metalic, a new approach to predicting properties of protein sequences. Metalic involves two phases: a "meta-learning" phase, during which a PLM backbone is fine-tuned to learn from data from multiple related fitness prediction tasks presented in-context, and then a per-task "fine-tuning" phase, where models are fine-tuned at inference time on a small amount of data specific to the task at hand. The authors show that Metalic outperforms baselines in the zero-shot setting for single mutations and is competitive for multiple mutations.

**Strengths:**

The paper is clearly written, includes a range of baselines and ablations, and also attacks an important problem. In some sense, protein fitness prediction is an excellent application for meta-learning, in that it's one of the few settings I can think of where there is an almost arbitrarily broad distribution of tasks that are nevertheless closely related to one another.

**Weaknesses:**

I'm a bit conflicted about this paper. On the one hand, the method seems to work, and, as I mentioned, the authors already include most of the experiments that I'd think to ask for. On the other: Table 5 shows that the lion's share of the improvement from Metalic comes from the in-context meta-learning; the per-task fine-tuning phase only helps in the many-shot case. But Metalic is only the best method in the zero-shot case (albeit with fewer parameters than baselines), where the only possible improvement comes from meta-learning. If the authors provided parameter-matched versions of Metalic that showed a much clearer benefit to using the method even in the many-shot case, that might change my thinking, but that also obviously changes the viability of performing the fine-tuning step at inference time. As it stands, the paper seems to show that in-context meta-learning works well  for zero-shot protein fitness prediction. In-context meta-learning isn't new, and I just don't know if a simple application of a method that already exists to a task that---for ICLR readers---is ultimately fairly niche warrants acceptance. I'm obviously a bit conflicted and can likely be convinced, but at the moment I'm leaning reject.

**Questions:**

While the paper is fairly clearly written, there are a few pieces of information I haven't been able to find:

1. Why did you stop at 8M? How much more computationally intensive would it be to run experiments with the 35M model?
2. How many tasks are presented in-context to generate the figures in e.g. Table 1? What data is used in-context at evaluation time? Is there overlap with the data used during the meta-learning phase?
3. "Moreover, our method still conditions on an unlabeled query set, and the protein embeddings in that query set, which allow for meta-learning a form of in-context unsupervised adaptation." -> Is there any evidence in the paper that this occurs? Would be cool to see an ablation of the data from other tasks to isolate this.

---

> ### Author Response · Authors · 2024-11-20
>
> Thank you for your feedback and for noting our comprehensive experiments. We appreciate your careful reading of our paper, and being open to discussion. We provide responses and clarifications below. Please kindly consider raising your score accordingly, or let us know if any topic remains unclear.
>
> **Weaknesses:**
>
> The primary concern seems to be with the lack of novelty of in-context meta-learning in the zero-shot setting, or the lack of improvement in the many-shot (n=128) setting. We address this concern here.
>
> While the largest gains from per-task fine-tuning appear in the many-shot setting (n=128), we actually saw significant improvement in the few-shot setting as well (n=16). For example, we ran this experiment and saw that fine-tuning improved the Spearman correlation from .476 (± .004) to .490 (± .003) in the single-mutant setting. The fact that the meta-learning helps more than the fine-tuning as n decreases is not surprising: The more data is limited, the more meta-learning prior beliefs from related data is useful.
>
> Moreover, it is not the case that fine-tuning does not help in the zero-shot setting; rather, it is simply not possible to use fine-tuning in that setting, since no additional sequences are available for it to be applicable. However, we believe that we have clearly demonstrated that the novel combination of in-context meta-learning followed by fine-tuning is highly effective, and that in-context meta-learning alone can be highly effective, even in the zero-shot setting, which is a surprising and informative result unto itself.
>
> Finally, we show that our method outperforms alternative forms of in-context meta-learning and gradient-based meta-learning, even in the many-shot (n=128) setting. This suggests that, while beyond the scope of this study, Metalic can be used more widely than just for protein fitness prediction in the future. (We believe the main reason that Metalic is not state-of-the-art in the many-shot setting has more to do with implementation details and scale than methodology.)
>
> We believe that our method has been demonstrated to be highly effective in pushing the state-of-the-art for zero-shot and few-shot protein fitness prediction, has a novel combination of in-context meta-learning and fine-tuning that helps in both the few-shot and many-shot settings (n=16 and n=128), and outperforms alternative meta-learning methods even in the many-shot setting. For these reasons we believe the work will have a high impact both in meta-learning and in low-data protein fitness prediction communities.
>
> **Questions:**
>
> 1. **“Why did you stop at 8M?”** We too had this question. We stopped at 8M because the model started to become impossible to fit on a single GPU with significantly more parameters. We did evaluate the 35M model, in response to this request, in the 128-shot setting, for 35,000 steps. The Spearman increased from 0.541 (± 0.004) to 0.550 (± 0.001), suggesting there is some gain possible, but that the improvement is modest. We hypothesize the reason the gain is not larger is that the meta-learning stage drives most of the improvements from our method, and there is not yet sufficient data in the meta-learning stage to take advantage of much larger models. We expect this will change as ProteinGym grows and more data becomes publicly available. Thank you for the question!
> 2. **“How many tasks are presented in-context to generate the figures… Is there overlap with the data used during the meta-learning phase?”** We evaluate over eight held-out single-mutant tasks (and five held-out multi-mutant tasks). There are 189 tasks total. Since the eight single-mutant (and five multi-mutant) tasks are held-out from meta-training, there is no overlap with the data used in the meta-learning phase. The data used in-context at evaluation time consists of a support and query set. These sets are subsampled from the specific task that is being evaluated, one task at a time.
> 3. **“unsupervised adaptation." -> Is there any evidence in the paper that this occurs?”** Thanks for asking! Yes, there is! Since we evaluate on held-out tasks, when we evaluate the zero-shot setting, the model has never seen any labels for the given task. We dive into this phenomenon in Section 5, where we show that even in the zero-shot setting, the model is attending to other proteins in the context. This means that other proteins are useful, despite having no associated labels ever given to the model! We found this phenomenon to be fascinating and highly useful. Future work could explore the mechanisms enabling this capability.

---

> > ### Comment · Reviewer_ZGWy · 2024-11-24
> >
> > I thank the authors for their hard work and am satisfied by the answers to my questions. Ultimately, I'm still not sold on the novelty of the paper (at least for a generalist conference like ICLR): the authors claim "We believe the main reason that Metalic is not state-of-the-art in the many-shot setting has more to do with implementation details and scale than methodology," and that might be the case, but the strongest results in the current paper by far are simply an application of meta-learning to protein tasks.

---

> > > ### Author Response · Authors · 2024-11-24
> > >
> > > Thank you again for your feedback. Realizing your concern is that our paper appears to be “just an application of meta-learning” for a “generalist conference like ICLR,” we would like to emphasize two key points that we hope will be persuasive:
> > >
> > > 1. **Alignment with ICLR’s Scope:** ICLR explicitly invites papers on “applications to physical sciences (physics, chemistry, biology, etc.)” in its call for papers. We believe that our work, situated at the intersection of machine learning and protein sciences, aligns directly with this criterion.
> > > 2. **Methodological Novelty:** We believe that the combination of in-context meta-learning followed by fine-tuning, and the type of fine-tuning proposed, is a novel contribution to the meta-learning field. Traditional approaches typically rely on either in-context meta-learning (learning to in-context learn) or gradient-based meta-learning (learning to fine-tune) independently (Beck et al., 2023). By treating in-context meta-learning as an initialization for fine-tuning, we address limitations inherent in these standalone methods. Furthermore, we introduce a novel fine-tuning procedure to mitigate memorization of the query set—a problem unique to combining in-context learning with fine-tuning. In fact, our research on proteins was bottlenecked precisely because we had been trying the typical approaches. Discovering our method to overcome these challenges required substantial effort and computational resources, and we believe it will benefit future researchers across domains, not just in protein sciences.
> > >
> > > Please do let us know your thoughts on these two points. We hope you might reconsider the value of this work, and we believe this added context highlights the dual significance of our work as both a meaningful application, in line with ICLR’s call for contributions, and as a methodological contribution to the field of meta-learning.

---

> > > > ### Comment · Reviewer_ZGWy · 2024-11-24
> > > >
> > > > 1. I fully agree that protein papers are just as welcome at ICLR as, say, vision papers. That being said, a simple application of what I'm purporting is effectively not very different from existing methods to a new domain is not automatically novel.
> > > > 2. It's clear that the paper is the result of substantial effort and computational resources, but I still haven't seen anything to refute my original point that the most impressive results in the paper occur in the setting where Metalic is exactly identical to regular meta-learning approaches.

---

> ### Author Response · Authors · 2024-11-25
>
> Thanks for your response. We would like to highlight three points:
>
> 1. **Contribution in the 0-Shot Setting:** Regular meta-learning methods do not typically address the 0-shot setting in supervised learning. In contrast, our work demonstrates the surprising emergence of unsupervised adaptation in this setting, which we believe is a significant novel contribution on its own.
> 2. **Strong Improvements Beyond 0-Shot:** The novel components of Metalic improve performance in other settings as well (16-shot and 128-shot). Metalic is also the strongest method in the 16-shot setting, which is only possible due to our contribution.
> 3. **Reasonable Interpretation of Guidelines:** Interpreting the ICLR guidelines to mean that any novel application (which we provide) must also introduce a novel method (which we also provide) and that the greatest improvement from that novelty must occur precisely where the method is strongest, seems like an unreasonably high standard that is unlikely to align with the intent of the ICLR guidelines.
>
> We truly appreciate your thoughtful consideration of our work, and we hope these clarifications demonstrate the significance and broad impact of our contributions. We would be grateful if you might revisit your assessment in light of this added context.

---

### Official Review · Reviewer_JKXe · 2024-10-28

**Soundness:** 3
**Presentation:** 3
**Contribution:** 3
**Rating:** 6
**Confidence:** 4

**Summary:**

The paper titled introduces Metalic, a meta-learning approach for predicting protein fitness, which is a measure of protein properties like stability and binding affinity. Predicting these properties accurately is challenging due to the scarcity of labeled data. Metalic improves on this by applying meta-learning to a distribution of related protein fitness tasks, enabling positive transfer to unseen tasks. The approach combines in-context meta-learning and fine-tuning without accounting for fine-tuning during meta-training, which helps to generalize well even with limited data. Experiments show that Metalic achieves state-of-the-art performance in zero-shot and few-shot settings on the ProteinGym benchmark, using 18 times fewer parameters than current models.

**Strengths:**

1-By integrating in-context meta-learning with subsequent fine-tuning, Metalic can generalize well to new tasks, even though fine-tuning is not accounted for during meta-training. Moreover, the model is more computationally efficient than state-of-the-art approaches.

2-Metalic achieves strong performance in zero-shot and few-shot protein fitness prediction tasks, which is particularly valuable given the limited availability of labeled data for many protein-related tasks.

**Weaknesses:**

1-I anticipate a more in-depth analysis of the correlation between PLM probability and fitness values in the context of meta-learning, especially when the authors assume that this correlation may not be reliable.

2-Although in-context learning is a strength, its effectiveness diminishes for tasks that deviate significantly from the distribution of tasks used during meta-training, potentially limiting generalization to out-of-distribution tasks. Experimental results on some OOD tasks can help readers better understand the paper.

3-The paper primarily compares Metalic to a few gradient-based meta-learning methods (e.g., Reptile) but does not extensively benchmark against other recent meta-learning approaches that incorporate different strategies for handling fine-tuning or leveraging unsupervised data.

4-Metalic shows strong performance on single-mutant tasks but falls short of the state-of-the-art on multi-mutant tasks. The authors attribute this to the limited availability of multi-mutant training data, emphasizing the need for more diverse datasets to fully realize the method's potential. A more in-depth analysis of this limitation would be valuable.

5-While the ProteinGym benchmark is a standard for protein fitness prediction, additional evaluations on other benchmarks would provide a more comprehensive understanding of Metalic's strengths and weaknesses across a broader range of protein-related tasks.

**Questions:**

1-The authors assume that the correlation between PLM likelihood and protein fitness may not be reliable. How does Metalic perform when the assumed correlation is weaker? Would additional analysis on this correlation help refine the model's approach?

2-In-context learning relies heavily on the distribution of tasks used during meta-training. How does Metalic handle tasks that are significantly different from the meta-training distribution, and are there strategies that could improve its generalization to such out-of-distribution tasks?

3-Given that Metalic does not account for fine-tuning during meta-training, how does this omission affect the model's ability to adapt to tasks with extensive fine-tuning requirements? Would incorporating some level of fine-tuning awareness during meta-training improve results?

---

> ### Author Response · Authors · 2024-11-20
>
> Thank you for the review, the detailed feedback, and for noting the strong performance of our method. We respond to your comments below. Please do let us know if we have addressed your concerns, and please do increase the score correspondingly if we have. Thank you for your help.
>
> **Weaknesses:**
>
> 1. **“I anticipate a more in-depth analysis of the correlation between PLM probability and fitness values in the context of meta-learning, especially when the authors assume that this correlation may not be reliable.”** We believe there may be some miscommunication here. Rather than assuming the correlation may not be reliable, we actually do measure this correlation. Specifically, each of our baseline methods, in the zero-shot setting, use the protein likelihood directly as the fitness prediction. The results given in the zero-shot setting are precisely the Spearman correlation for PLM probabilities. Note that the fact that all baselines must rely on the PLM probabilities is the exact reason that Metalic is able to outperform them in the zero-shot setting. Using the additional meta-learning stage, ours is the only method that is able to learn how to use the PLM probabilities rather than directly using them for the zero-shot setting.
> 2. **“Although in-context learning is a strength, its effectiveness diminishes for tasks that deviate significantly from the distribution of tasks used during meta-training, potentially limiting generalization to out-of-distribution tasks.”** All of our tasks are non-parametric and are held-out from the training set, and are thus OOD tasks. It is true that in-context learning can struggle here, which is precisely the reason that the combination with fine-tuning in Metalic is so effective.
> 3. **“Does not extensively benchmark against other recent meta-learning approaches that incorporate different strategies for handling fine-tuning or leveraging unsupervised data.”** We compare to Reptile, since it is a widely used method for making gradient-based methods more efficient, which is necessary given the computational demands. To leverage unsupervised data in the query set, we combine Reptile with Metalic in Metalic-Reptile, which can additionally condition on the unsupervised data in-context, and still find it not worth the extra computation. Are there other methods for unsupervised meta-learning that you had in mind and expect to work well here?
> 4. **“Metalic shows strong performance on single-mutant tasks but falls short of the state-of-the-art on multi-mutant tasks … a more in-depth analysis of this limitation would be valuable.”** We have added an additional plot of the performance as we change the number of multi-mutant training tasks, both with and without the single-mutant tasks. This additional analysis allows us to see that more multi-mutant data is useful in either case, strengthening our analysis and argument. The plot is available in the updated manuscript
> 5. **“additional evaluations on other benchmarks would provide a more comprehensive understanding of Metalic's strengths”** We use ProteinGym since it has become the standard aggregated benchmark in the field. Note that ProteinGym is comprehensive and is itself composed of roughly 200 individual tasks, each of which is its own dataset collected by different researchers. ProteinGym aggregates available tasks, and thus spans many relevant protein-related tasks. We remain open to suggestions for tasks that are not included in ProteinGym and would be beneficial for evaluation, but do not know of any such standard benchmarks.
>
>
> **Questions:**
>
> 1. **“The authors assume that the correlation between PLM likelihood and protein fitness may not be reliable. How does Metalic perform when the assumed correlation is weaker?”** Please see answer 1) in the Weakness section.
> 2. **“In-context learning relies heavily on the distribution of tasks used during meta-training. … are there strategies that could improve its generalization to such out-of-distribution tasks?"** Please see answer 2) in the Weakness section.
> 3. **“Given that Metalic does not account for fine-tuning during meta-training, how does this omission affect the model's ability to adapt to tasks with extensive fine-tuning requirements?”** This is a great question, and one that we investigate extensively in Section 4.6. Specifically, we evaluate Metalic-Reptile, to account for the fine-tuning during meta-training. Metalic-Reptile is equivalent to Metalic, but adds the objective from Reptile that accounts for fine-tuning during meta-training. It turns out that even when fine-tuning, the improvement from Metalic-Reptile is marginal, and the increased computation is large. When controlling for the total number of gradient computations, Metalic-Reptile increases implementation complexity with no clear advantage.

---

> > ### Comment · Reviewer_JKXe · 2024-11-25
> > **Thank you for the rebuttal**
> >
> > I've read the rebuttal and thank the authors for their efforts.

---

### Official Review · Reviewer_Q2KC · 2024-11-04

**Soundness:** 3
**Presentation:** 3
**Contribution:** 3
**Rating:** 6
**Confidence:** 3

**Summary:**

This paper introduces Metallic, which performs few-shot (N=16 or N=128) or zero-shot fitness prediction using in-context meta-learning, using the ProteinGym benchmark. The model is meta-trained across many protein prediction tasks, to enable in-context learning, then finetuned on the support set, where the support set defines the number of "shots" for the few-shot task. Evaluations uses pairwise ranks (i.e. if model correctly prefers one mutant to the other). The meta-training procedure yields models that fare better in the finetuned few-shot setting than baselines (ESM1, etc.)

**Strengths:**

* The idea is relatively under-explored in protein ML despite ICL having shown strong performance in other fields of ML, and the importance of few-shot learning.
* Paper is fairly clearly written
* Attention map analysis is a great addition to the work! Would love to see follow up on this.

**Weaknesses:**

* In Table 1, 2, and 3a, only ESM2-8M is used for comparison. What happens if we compare to the larger ESM2 models? Can we get better zero-shot or few-shot performance via scaling pretraining parameters?
* Results in Figure 3b and 3a don't seem too much better than baselines.
* More rigorous examination of the effects of varying query set size, etc. might be quite insightful
* Some papers have pointed out that protein likelihoods don't always indicate protein fitness, and might be due to the training data biases [1,2].

Minor / meta comment:
I think that though thinking about ICL for proteins is a good idea, and authors demonstrate thoughtfulness in how this should be done. However, there might be more interesting tasks to examine than likelihood & protein fitness. There has also been working showing that "evotuning" (finetuning on evolutionary neighbors) can improve performance [2,3]. In a real-world setting, if I have a protein for which my pretrained model is underperforming on, I'm not sure if I would prefer to use a meta-trained model or a evo-tuned model.

From a writing perspective, authors might be able to increase impact by outlining real-world drug discovery scenarios where this is relevant.

[1] Protein language models are biased by unequal sequence
sampling across the tree of life. Ding et al., 2024
[2] Protein Language Model Fitness Is a Matter of Preference. Gordon et al., 2024
[3] Unified rational protein engineering with sequence-based deep representation learning. Alley et al., 2024

**Questions:**

* Are there some tasks that are easier to learn vs. harder to learn? It would be interesting to see a breakdown analysis.
* Do authors have hypotheses for why the performance gap between Metalic and baselines is so much more pronounced for zero-shot vs. few shot tasks?

---

> ### Author Response · Authors · 2024-11-20
>
> Thank you very much for your detailed feedback. We appreciate the review and thank you for noting the importance of the application area and the value of the analysis. Responses to your comments are given below. If we have sufficiently addressed your concerns, we kindly ask that you raise your score accordingly, or let us know if further clarification is needed.
>
> **Weaknesses:**
>
> 1. **“Can we get better zero-shot or few-shot performance via scaling pretraining parameters?”** We did evaluate the 35M model, in response to this request, in the 128-shot setting, for 35,000 steps. The Spearman increased from 0.541 (± 0.004) to 0.550 (± 0.001), suggesting there is some gain possible from scaling the model, though the improvement is modest. We hypothesize the reason the gain is not larger is that our meta-learning stage drives most of the improvement over the baselines, and there is not yet sufficient data in the meta-learning stage to take advantage of much larger models. We expect this will change as ProteinGym grows and more data becomes publicly available.
> 2. **“Results in Figure 3b and 3a don't seem too much better than baselines.”** The multi-mutant results are indeed outperformed by ESM-IF1 and PoET; however, as we hypothesize in the manuscript, this limitation is likely due to available data and not due to methodological limitations. We provide evidence of positive scaling with the number of tasks in Table 4, and we have added an additional graph into the manuscript to depict this scaling. Moreover, our method is uniquely placed to take advantage of any newly released dataset in the future, since it is the only method that leverages a meta-training stage. This means that our method is the only method that will actually improve with more datasets released over time.
> 3. **“More rigorous examination of the effects of varying query set size, etc. might be quite insightful.”** We did perform some experiments to evaluate this, not reported in the paper, since they used older hyper-parameters. Primarily, we saw that in the zero-shot setting, performance did increase marginally from 0.461 (± 0.004) to 0.471 (± 0.005), when increasing the query size from 100 to 200. We could not increase the query size much more than this due to computational constraints. We also note that forming firm hypotheses from experiments on the query size is quite difficult, since the larger the query size the less variance there is in the gradient estimation.
> 4. **“Some papers have pointed out that protein likelihoods don't always indicate protein fitness, and might be due to the training data biases.”** This is exactly the reason that metallic is able to outperform the baselines. The baselines make this assumption in the zero-shot setting, whereas the meta-learning phase enables Metalic to NOT rely on likelihoods to predict fitness.
> 5. **“Minor / meta comment…There has also been working showing that "evotuning" (finetuning on evolutionary neighbors)”** We believe that fine-tuning on evolutionary neighbors could be used in conjunction with our method, in the zero-shot setting, not instead of it. Additionally, the ESM pre-training stage may already give our model a similar benefit, since it has had access to similar homologs in pre-training.
> 6. **“From a writing perspective, authors might be able to increase impact by outlining real-world drug discovery scenarios where this is relevant.”** We have added text to the introduction to clarify this point, discussing how protein fitness prediction can be used to optimize properties such as the binding affinity of a monoclonal antibody therapy to its target, or the thermostability of enzymes functioning at high temperatures.
>
> **Questions:**
>
> 1. **“Are there some tasks that are easier to learn vs. harder to learn?”** Certainly some of the tasks are easier and some are harder. For example, in the 128-shot single mutant evaluation, the Spearman correlation for Metalic ranged from 0.139 (± 0.009) to 0.852 (± 0.020), with a median of 0.578. (The mean Spearman correlations, in order, are .139, .342, .445, .547, .610, .688, .794, .852.) The large range in Spearman correlation shows a broad range of difficulties for evaluation tasks.
> 2. **“Do authors have hypotheses for why the performance gap between Metalic and baselines is so much more pronounced for zero-shot vs. few shot tasks?”** Yes, the result was not surprising to us given the purpose of meta-learning. Meta-learning adds an additional training stage to learn prior beliefs and inductive biases from related data. The more limited the data is, the more useful it is to rely on prior data. We have made a note of this in Section 4.3.

---

> > ### Comment · Reviewer_Q2KC · 2024-11-26
> >
> > Thanks for the response to these comments and the additional experiments. A few more questions:
> >
> > * _"The multi-mutant results are indeed outperformed by ESM-IF1 and PoET; however, as we hypothesize in the manuscript, this limitation is likely due to available data and not due to methodological limitations."_ Could authors expand more on this / point me to the lines in the manuscript where this is explained? I'm not sure if I grasp at first glance why this should be the case; shouldn't the methods have access to the same downstream data? Or do authors mean that this is due to pretraining data?
> > * The authors have addressed my questions about likelihoods & evotuning, and how meta-learning allows us to move away from sometimes unreliable likelihoods in the zero-shot setting.
> > * Re: introduction, thank you for making this update. (My point was more about why specifically _few-shot_ or _zero-shot_ learning is useful; for example, what are situations where we might want to make a zero-shot prediction? if we already have the assay to perform a few experiments for a few setting, why don't we just perform the rest of the assay? I'm not asserting that few-shot/zero-shot is not useful, but I think contributing more precise and critical thinking on why _few-shot_ and/or _zero-shot_ is specifically useful for real-world practitioners who seem themselves in that description would strengthen the utility of this paper)
> > * Re: tasks, my question was intended to be about _why_ some tasks are easier to learn than others. What does that tell us about the tasks? I think that if the authors can make a rigorous inquiry on this front, it would make for a much more impactful paper.
> >
> > Overall, I'm not sure if these experiments are able to pull my score up to a confident Accept (if 7 was an option, I'd probably raise my score to that!) Few-shot and zero-shot learning for protein fitness prediction problem is an important problem, and this paper gets most of the details right, but there are a few critical pieces that feel a bit under-developed, and feels a bit like throwing meta-learning literature wholesale onto proteins. I think this work is still borderline, but I do think that the ICLR community would find this work to be helpful inspiration.

---

> ### Author Response · Authors · 2024-11-27
>
> **Thank you for your thoughtful questions and detailed engagement with our paper. We address each point below, with specific attention to the applicability of zero-shot and few-shot settings, informed by discussions with our protein engineering domain expert. We hope these answers will help increase your confidence in our work.**
>
> 1. **Multi-Mutant Data:** The limitation in performance is likely due to the difference in available datasets: We have 121 single-mutant datasets but only 63 multi-mutant datasets for meta-training (pre-training). We explore this hypothesis in Section 4.4, where we demonstrate that performance improves significantly with additional meta-training data, suggesting that the bottleneck is indeed the lack of sufficient multi-mutant data.
> 2. **Real-World Application of Zero-Shot and Few-Shot Settings:** Protein engineering often involves mutating residues to test or alter an aspect of fitness. To clarify the utility of zero-shot and few-shot predictions, we provide a practical example: Suppose a research lab discovers a monoclonal antibody with both reactivity and neutralization potential against a broad panel of viral strains. To investigate whether this monoclonal should be developed as a therapeutic, researchers may want to identify mutations within the binding pocket that could lead to viral escape. For a particular residue of interest, the researcher may need to mutate that residue to all 20 possible amino acids, requiring the design and expression of up to 19 additional protein variants—a significant investment of time and resources. Few-shot prediction models can save substantial effort and cost. For example, the researcher could assay fitness for 10 variants experimentally and use a few-shot prediction model to predict the fitness of the remaining 10 variants, reducing labor by half. This scenario corresponds to a 10-shot fitness prediction problem. Predictions improve with additional data, so incorporating variants from other residues in the binding pocket would further enhance accuracy. For instance, if escape profiles were already known (from previously published work) for 10 individual positions, predicting the escape profile of an 11th position becomes a ~200-shot prediction problem. (Note that a researcher could generate the entirety of this dataset themselves using a Deep Mutational Scanning Library. However, depending on the type of scanning library made, there may be a large upfront cost and the costs could scale significantly with the number of residues mutated or the length of the protein. So, it may be preferable to avoid a DMS altogether, or it could make sense to generate only a subset of these 11 residues within the library and then predict the rest.)  Zero-shot prediction is even more impactful, as it eliminates the need for prior assays altogether, avoiding all upfront costs. While not all tasks lend themselves to few-shot or zero-shot prediction, many do (e.g., mutations sharing biophysical side-chain properties, such as electrostatic charge). Empirically, this is supported by our up to 85% Spearman correlation on held-out validation tasks. These scenarios were developed in consultation with a protein engineering domain expert actively conducting wet lab research, underscoring their relevance to real-world practice.
> 3. **Task Difficulty:**  Differences in the amino acid sequence and the property being assayed produce distinct challenges for modeling. The tasks are derived from Notin et al. (2023), who note that they “cover diverse protein families in terms of taxa, mutation depths, sequence lengths, and MSA depth.” Additionally, the properties of these proteins vary significantly. Both the protein sequence and the assessed property will affect difficulty. For example, in tasks that involve predicting epistatic effects on viral escape, protein fitness will be a highly complex function of the protein sequence. In contrast, in tasks such as direct binding interactions, knowing the effects of one mutation gives considerable information about neighboring mutations, resulting in an easier protein fitness prediction task. Additionally, as mentioned in answer 1), there are known issues due to data constraints in the multi-mutant setting. While we understand the reason that some of these tasks are difficult, either due to data constraints or more complex relationships between the protein sequence and its associated fitness, systematically varying evaluation data by the protein sequences and properties would be a useful feature in a standard benchmark, and such an effort would represent a valuable future contribution.
>
> **Thank you for noting that you believe the ICLR community would benefit from this paper. We hope these responses address your concerns and further increase your confidence in our work. We greatly appreciate your engagement and detailed feedback.**

---

### Official Review · Reviewer_suA1 · 2024-11-04

**Soundness:** 3
**Presentation:** 3
**Contribution:** 3
**Rating:** 8
**Confidence:** 4

**Summary:**

This work introduces Metalic, a framework for performing protein sequence fitness prediction in both in-context and few-shot regimes. It proposes to use in-context-learning as a replacement for gradient updates used in the traditional inner loop of meta-learning. While using the smallest ESM-2 model, it achieves strong performance on the chosen fitness tasks.

**Strengths:**

Metalic achieves success in both providing a compute efficient alternative to meta-learning and also provides strong performance on downstream tasks. It’s also noteworthy the parameter efficiency that Metalic achieves in the process.

The work is also highly reproducible. Each experiment could be reproduced by a reader from the text alone (barring which datasets were and were not used). In the appendix, readers can find a comprehensive set of hyperparameters and training details used in each experiment.

When comparing metalic, it uses relevant choices in baseline models and also analyzes where and how performance degrades (e.g. the case of multi-mutants). In addition, the ablations show the utility of the different components of the method, even displaying its ability to outperform traditional meta-learning methods like Reptile. The experiments are very thorough in justifying all choices made during modeling.

**Weaknesses:**

The only major weakness that appears in this work is the limited number of datasets used in evaluating the proposed methods. ProteinGym performance fluctuates a lot amongst tasks, hence it would be nice to decouple the comparatively small number of datasets, 13 in total, for evaluation from performance. A simple experiment selecting different splits and training replicates in a similar manner to a partial k-fold cross validation would be convincing.

On a more minor note, the exact implementation of the algorithm would benefit from a slightly more detailed treatment. Though it can be inferred from Figure 2a, it would benefit from some support either pictorially or mathematically in the exact computations that must be performed.

**Questions:**

* How does performance change when training and evaluating on a separate set of proteins?
* How do inference and training costs of Metalic compare to the Reptile ablations?
* To clarify the architecture, would it be possible to have a Vaswani et al. 2017 style figure in the appendix of the exact computations that occur?

Overall the work is quite strong, and would benefit from a simple experiment to further prove its robustness.

Vaswani et al. 2017, "Attention Is All You Need"

---

> ### Author Response · Authors · 2024-11-20
>
> Thank you very much for your review and feedback. Thank you for noting that the performance is strong and that the experiments are thorough. We respond to the weaknesses and questions below.
>
> **Weaknesses:**
>
> **“This work is the limited number of datasets used in evaluating… a partial k-fold cross validation would be convincing”** We agree that evaluating over diverse datasets is an important consideration. While a k-fold evaluation would be prohibitively expensive (approximately 192 days per seed with each evaluation over eight datasets), we do take care to supplement with multi-mutant evaluation and ensure that the single-task datasets match prior work and are diverse in terms of data and difficulty. The original justification, taken from Notin et al. (2023), is that “together, these 8 assays cover diverse protein families in terms of taxa, mutation depths, sequence lengths and MSA depth.” Moreover, the Spearman correlation of our models ranges from 0.139 to 0.852 over the 8 datasets, further suggesting a broad range of difficulty within our evaluation set.
>
> **“On a more minor note, the exact implementation of the algorithm would benefit from a slightly more detailed treatment”** Thanks for noting the difficulty! To make the exact computation clear, we have added a figure with more architectural detail to the appendix. (Note that we will also release the accompanying codebase publicly upon acceptance to ensure that the exact computations are clear and reproducible.)
>
> **Questions:**
>
> 1. **“How does performance change when training and evaluating on a separate set of proteins?”** We chose the evaluation set to align with prior work (Hawkins-Hooker et al., 2024 and Notin et al. 2023), and do not have the computational resources to run a k-fold evaluation, but appreciate the question.
> 2. **“How do inference and training costs of Metalic compare to the Reptile ablations?”** Reptile inference costs are similar, but the training is significantly more expensive. For every gradient update of Metalic, Reptile must use three times the number of gradient computations. (Note that if we wanted conditions to match exactly, Reptile would use 100 times the number of gradient computations, but this was impossible due to compute limitations.) For this reason, we originally compared to Reptile after training for only 15,000 steps. However, we have run more experiments for the rebuttal, and the Reptile experiments have now been trained for the entire 50,000 steps. We have updated the manuscript with these results.
> 3. **“Vaswani et al. 2017 style figure in the appendix of the exact computations that occur?”** Thank you for the suggestion, we have now added such a figure to the appendix in response. Note that we will also release the codebase to ensure reproducibility!

---

> > ### Comment · Reviewer_suA1 · 2024-11-26
> > **Response to Comment by suA1**
> >
> > I'd like to thank the authors for their time spent addressing the points raised above. The reasoning behind not having a full k-fold validation is understandable. With that being said, the work could still be improved by for example rerunning the method another 4 times to get a basic estimate of the variation. Reviewing the figure added to the appendix, the algorithm is now easy to follow improving the reading of the paper. I believe my current score reflects the strength of the paper, so it will remain.

---

### Meta-Review · Area_Chair_pTpm · 2024-12-19

**Metareview:**

The authors propose Metalic (Meta-Learning In-Context), a meta-learning framework for protein fitness prediction. By combining in-context learning with fine-tuning, Metalic adapts to new tasks with fewer parameters and better generalization. It outperforms state-of-the-art models in low-data settings, setting a new benchmark on ProteinGym. This work demonstrates the potential of meta-learning in advancing protein engineering, particularly in data-scarce scenarios.

**Strengths:**

- The integration of in-context meta-learning with fine-tuning is novel and computationally efficient compared to state-of-the-art approaches.
- The paper is clearly written and highly reproducible.
- **Metalic** demonstrates strong performance in zero-shot and few-shot protein fitness prediction tasks, which is especially valuable given the scarcity of labeled data in protein-related fields.

**Weaknesses:**

- More experiments and analyses are needed to strengthen the findings.
- The novelty of the approach should be explained in greater detail.

**Overall:** This paper addresses a significant problem in protein learning. While the use of meta-learning in this domain is not entirely new, the specific combination of in-context meta-learning followed by fine-tuning, along with the proposed fine-tuning approach, represents a novel and valuable contribution. Although the results do not achieve state-of-the-art performance across all tasks, they are strong enough to support the authors' claims. I recommend acceptance.

**Additional Comments On Reviewer Discussion:**

During the rebuttal period, the authors addressed the following points:

- In response to Reviewer suA1, Q2KC, and JKXe, the authors provided additional experiments and explanations to address concerns regarding experimental settings, dataset construction, ablation studies, and experimental analyses. All three reviewers gave positive scores in the end.

- Reviewer ZGWy raised concerns about the novelty of the methodology and the significance of the experimental results. The authors clarified that the combination of in-context meta-learning followed by fine-tuning, along with the proposed fine-tuning approach, represents a novel contribution to the meta-learning field. They also demonstrated that **Metalic** improves performance in other settings as well (16-shot and 128-shot). After reviewing the revised paper and the comments from both the reviewers and authors, I agree that this paper provides valuable contributions to the domain.

---

### Decision · Program_Chairs · 2025-01-22

Accept (Poster)